# The Reentry Helix Is Potentially Involved in Cholesterol Sensing of the ABCG1 Transporter Protein

**DOI:** 10.3390/ijms232213744

**Published:** 2022-11-08

**Authors:** Zoltán Hegyi, Tamás Hegedűs, László Homolya

**Affiliations:** 1Institute of Enzymology, Research Centre for Natural Sciences, H-1117 Budapest, Hungary; 2Department of Biophysics and Radiation Biology, Semmelweis University, H-1094 Budapest, Hungary; 3ELKH-SE Biophysical Virology Research Group, Eötvös Loránd Research Network, H-1094 Budapest, Hungary

**Keywords:** ABC transporter, ABCG1, ABCG4, CRAC motif, sterol binding element, apoptosis, protein structure

## Abstract

ABCG1 has been proposed to play a role in HDL-dependent cellular sterol regulation; however, details of the interaction between the transporter and its potential sterol substrates have not been revealed. In the present work, we explored the effect of numerous sterol compounds on the two isoforms of ABCG1 and ABCG4 and made efforts to identify the molecular motifs in ABCG1 that are involved in the interaction with cholesterol. The functional readouts used include ABCG1-mediated ATPase activity and ABCG1-induced apoptosis. We found that both ABCG1 isoforms and ABCG4 interact with several sterol compounds; however, they have selective sensitivities to sterols. Mutational analysis of potential cholesterol-interacting motifs in ABCG1 revealed altered ABCG1 functions when F571, L626, or Y586 were mutated. L430A and Y660A substitutions had no functional consequence, whereas Y655A completely abolished the ABCG1-mediated functions. Detailed structural analysis of ABCG1 demonstrated that the mutations modulating ABCG1 functions are positioned either in the so-called reentry helix (G-loop/TM5b,c) (Y586) or in its close proximity (F571 and L626). Cholesterol molecules resolved in the structure of ABCG1 are also located close to Y586. Based on the experimental observations and structural considerations, we propose an essential role for the reentry helix in cholesterol sensing in ABCG1.

## 1. Introduction

ABCG1 belongs to the G subfamily of ATP-binding cassette (ABC) transporters (recently reviewed in [1]). Unlike most other mammalian ABC transporters, these proteins consist of only one nucleotide binding domain (NBD) and one transmembrane domain (TMD); therefore, they are called ABC half-transporters. Another characteristic feature of the members of the G subfamily is the reverse domain order, meaning that, contrary to most ABC transporters, the NBD is localized at the N-terminus of the proteins. High-resolution atomic structures for various ABCG proteins, including ABCG5/ABCG8 [2], ABCG2 [3,4,5,6], and ABCG1 [7], have recently been published and collected into a database [8,9], revealing general structural homologies but also some dissimilarities among the members of this subfamily.

Several isoforms of ABCG1 have been reported, among which the variant consisting of 678 amino acids (aa) (G1-L) is studied the most since this isoform is primarily expressed in mice. In humans, however, the predominant isoform lacks a 12 aa-long segment between the NBD and TMD (G1-S) [10], and this variant also possesses a longer half-life than G1-L [11]. As regards its physiological function, ABCG1 has been proposed to play a role in HDL-dependent cellular lipid/sterol regulation, as cellular efflux of cholesterol and other lipid molecules to various acceptor molecules was associated with the functional expression of the transporter [10,12,13,14,15,16,17,18,19,20,21,22,23,24], and accumulation of various sterols was observed in the tissues/cells of Abcg1 single knockout and Abcg1/Abcg4 double knockout (or knock-in) animals [13,14,15,16,17,25]. A role in detoxification has also been proposed for ABCG1, protecting cells from toxic agents [26]. We have previously demonstrated that ABCG1 and the closely related ABCG4 induce apoptosis in cells [27,28]. Several sterol derivatives, including cholesterol, desmosterol, 7-ketocholesterol, and 7β -hydroxycholesterol, were identified as potential ABCG1 substrates on the basis of their efflux from ABCG1-expressing cells [12,14,15,16,24,26] (reviewed in [1]). However, whether ABCG1 actually transports these compounds or whether their cellular efflux is an indirect consequence of the functional presence of the transporter, it remains elusive.

Similarly, the regions in ABCG1 interacting with sterol compounds have not been revealed thus far. In other proteins involved in cellular sterol homeostasis, several specific amino acid sequences were identified as motifs responsible for protein-sterol interaction. These include the Cholesterol Recognition Amino acid Consensus (CRAC) and Sterol Binding Element (SBE) motifs [29,30]. The role of these motifs was previously investigated in ABCG2 to identify its cholesterol-interacting regions [31,32]. In the present work, we first screened for potential sterol substrates of the two major ABCG1 isoforms and ABCG4 and then investigated how mutations in the putative CRAC and SBE motifs in ABCG1 influenced the expression, cellular localization, and function of the transporter. The recently published atomic structure of ABCG1 also allowed us to interpret the functional consequences of the mutations in the context of the 3D structure of the transporter.

## 2. Materials and Methods

### 2.1. Insertion of the ABCG Protein Sequence into the pEGFP-N1 Vector and Expression in Mammalian Cells

The cDNAs of ABCG1 (NP_004906.3) and ABCG4 (NP_001135977.1) were cloned as described previously [27,28]. Briefly, the cDNA sequences were inserted next to the EGFP sequence in the pEGFP-N1 vector, leaving a stop codon and frame shift between EGFP and the encoded protein sequences. The shorter isoform (NP_058198.2) of ABCG1 and the different mutant variants of the short isoform were generated by site-directed mutagenesis. The following primer pairs (forward and reverse, respectively) were used as listed below:

ABCG1S:GGCCCTCTGAAGAGGACTCCTCGTCCATGGCCATGGACGAGGAGTCCTCTTCAGAGGGCCL430A:CCTCATTGGCCTGCTGTACGCGGGGATCGGGAACGAAGCCGGCTTCGTTCCCGATCCCCGCGTACAGCAGGCCAATGAGGL626A:CCAGAAGTCGGAGGCCATCGCACGGGAGCTGGACGTGGCCACGTCCAGCTCCCGTGCGATGGCCTCCGACTTCTGGY586A:CGTACCTACAGTGGATGTCCGCCATCTCCTATGTCAGGTATGGCCATACCTGACATAGGAGATGGCGGACATCCACTGTAGGTACGY655A:CCCTCCGCCTCATTGCCGCATTTGTCCTCAGGTACAAAATCCGCGGATTTTGTACCTGAGGACAAATGCGGCAATGAGGCGGAGGGY660A:GCCTATTTTGTCCTCAGGGCCAAAATCCGTGCAGAGAGGCCTCTCTGCACGGATTTTGGCCCTGAGGACAAAATAGGCF571A:GCTCCTGTTCTCGGGGTTCGCCGTCAGCTTCGACACCATCCGGATGGTGTCGAAGCTGACGGCGAACCCCGAGAACAGGAGC

The cDNAs of ABCG1 variants were sequenced to confirm that the target mutations are present in the specific vector.

The HEK293 and HeLa cells (ATCC) were maintained in complete DMEM in a humidified CO_2_ incubator at 37 °C. For transfection, 4 × 10^5^, 4 × 10^4^ or 2 × 10^4^ cells were seeded into 6-well (Greiner), 8-well (Nunc) Lab-Tek II chambers, or 24-well plates (Greiner), respectively. Twenty-four h after seeding, the cells were transfected with different constructs using a 1:4 ratio of DNA (µg) and Fugene HD reagent (µL) (Promega).

### 2.2. Expression of ABCG1 and ABCG4 Variants in Insect Cells; Assessment of Their ATPase Activity

The cDNA sequences of human ABCG1 (NP_004906.3) and ABCG4 (NP_001135977.1) were inserted into the pAcUW21-L expression vector as previously described [33]. Briefly, the 2038 nucleotide long cDNA of the long isoform of ABCG1 was amplified with primers ABCG1F (5′-caccatggcctgtctgatggccgc-3′) and ABCG1R (5′-tcctctctgcccggattttgtac-3′) by RT-PCR from macrophages. Next, the PCR fragment was inserted into the pcDNA3.1/CT-GFP-TOPO vector (Invitrogen) by TA-cloning. Subsequent PCR subcloning placed the cDNA in the baculovirus expression vector, pAcUW21-L, and added a stop codon. ABCG4 cDNA was purchased from the I.M.A.G.E. consortium (clone ID 1537140) and cloned into pAcUW21-L. ABCG1 and ABCG4 cDNAs were sequenced to confirm correct cloning. The short isoform and the mutant variants were generated in the pAcUW21-L vector as described above. Recombinant baculoviruses carrying transporter cDNA were generated with the flashBAC Ultra Kit (Oxford Expression Technologies) using Fugene HD reagent for infection. The proteins were expressed in the Sf9 (*Spodoptera frugiperda*) insect cell heterologous expression system. Infection and culturing of Sf9 cells, as well as membrane preparations, were carried out as previously described [34]. Various sterol compounds, including cholesterol, desmosterol, lathosterol, 7-ketocholesterol, 25-hydroxycholesterol, 27-hydroxycholesterol, β-sitosterol, and campesterol, were purchased from Sigma-Aldrich. Their complexation in randomly methylated β-cyclodextrin (CD) was carried out in Cyclolab, Ldt, as a kind gift from Lajos Szente. Membrane preparations containing the ABCG proteins or β-galactosidase (β-gal) were incubated with the CD-complexed sterol compounds at the indicated concentrations for 10 min. The vanadate-sensitive ATPase activity of ABCG proteins was measured by colorimetric detection of inorganic phosphate liberation as described previously [33].

### 2.3. Immunoblot Experiments

Western blot analysis of protein samples was performed as described previously [34] with some modifications. Briefly, the mammalian cells were scraped and suspended in 1× Laemmli buffer and sonicated three times for 10 s at 4 °C. The indicated amount of samples were loaded onto a Laemmli-type 7.5% SDS-polyacrylamide gel and separated. Membrane preparations of Sf9 cells were diluted in Laemmli buffer, and 100 ng of proteins were separated. The transfer of the proteins onto PVDF membrane was carried out in a Tris buffer containing a high concentration of glycine and 4% methanol at 200 mA for 1.5 h. Immunodetection was performed using a specific monoclonal antibody against ABCG1 developed in our laboratory [27]. The antibody was diluted 500- or 2000-fold in TBS-Tween-milk buffer for mammalian and Sf9 samples, respectively. HRP-conjugated goat anti-mouse IgG secondary antibody (Jackson Immunoresearch, 1:10,000) and enhanced chemiluminescence technique were applied to detect specific bands. In the case of mammalian samples, a mouse anti-Na^+^K^+^ ATPase antibody (Biomol, 1:1600) was used for loading control.

### 2.4. Immunofluorescence Staining

Immunofluorescence staining was carried out as described previously [35] with minor modifications. Briefly, 24 h after transfection, HEK293 and HeLa cells were washed in Dulbecco’s modified PBS. Fixation and permeabilisation were carried out with 4% paraformaldehyde and pre-chilled methanol. Fixation was followed by incubation for 1 h at RT in blocking buffer (0.2% BSA, 1% fish gelatin, 0.1% TX-100, and 5% goat serum). The samples were incubated with our anti-ABCG1 antibody (1:100 in blocking buffer) and either anti-pan-cadherin (Abcam, 1:100), anti-giantin (Abcam, 1:500), or anti-calnexin (Abcam, 1:100) antibodies for 1 h at RT. After a gentle washing step, the cells were incubated with species-specific, AlexaFluor-conjugated secondary antibodies (Life Technologies/Thermo Fisher Scientific) diluted in blocking buffer (1:250). Green (505–525 nm) and far-red (>660 nm) fluorescence images of the immunostained samples were acquired at 488 and 633 nm excitations, respectively. A Zeiss LSM710 confocal laser scanning microscope with a PLAPO 40× (1.4) oil immersion objective was used for image acquisition.

### 2.5. Detection of Apoptosis

For quantitative assessment of apoptotic cells, HEK293 and HeLa cells were seeded onto a 96-well plate and transiently transfected with the different forms of the short isoform of ABCG1. Twenty-four h later, the cells were incubated in Annexin V binding buffer containing 20-fold diluted Alexa Fluor 488-labeled Annexin V and 5 μM Hoechst33342 (Life Technologies/Thermo Fisher Scientific, Waltham, MA USA) for 5 min. Images were acquired by the ImageXpress Micro XLS wide-field high-content screening system (Molecular Devices) using a 10× Plan Fluor objective (Nikon, NA = 0.3). Twelve separate fields of view in each well were imaged for sampling the total cell culture. Using MetaXpress software, Annexin V-labeled cells and the total cell number were counted on the basis of green and blue fluorescence, respectively. Single cells were counted on the basis of Hoechst33342, a nuclear dye, with limits set at minimum and maximum width (10–25 µm) and minimum intensity difference from the background (1000 au). Next, Annexin V positive cells were identified, considering the nuclear staining to exclude cell debris without nuclear staining from the analysis. The results were expressed as the mean of a percentage of the total cell number ± S.E.M. For statistical analysis, a Student’s *t*-test was used to evaluate significant differences. (*p* < 0.05) in comparison with controls.

### 2.6. Structure Visualisation

The bottom-open apo structure of ABCG1 (PDBID: 2r8d) was used to analyse the spatial localisation of mutated residues. The PyMOL Molecular Graphics System, Version 2.4 (Schrödinger, LLC, New York, NY, USA), was employed for visualisation.

## 3. Results

### 3.1. The Effect of Sterol Compounds on the ATPase Activity of ABCG1 Isoforms

Stimulated ATPase activity is a widely used, sensitive method to monitor the interaction between various compounds and ABC transporters [34]. Substrate molecules regularly stimulate the basal ATPase activity of the transporter, whereas inhibitors diminish the substrate-stimulated ATPase activity. In several cases, the transported substrate also inhibits ATPase activity at high concentrations. This is well exemplified by the inhibitory effect of verapamil on the ATPase activity of MDR1/P-glycoprotein [34]. Although there is a strong correlation between molecules stimulating the ATPase activity and the transported substrates, it should be noted here that lack of stimulation of a molecule does not mean it is a substrate, and vice versa, having stimulatory effect is not evidence for being a substrate. The latter is exemplified by the allosteric effect of cholesterol on ABCG2 activity [36,37]. Nevertheless, direct interactions between compounds and the transporter can be studied using the ATPase activity assay. In a previous study, we monitored the interaction between the long isoform of ABCG1 and numerous substances [33]. Even though ABCG1 is connected with the cellular efflux of cholesterol, previous attempts to measure its stimulatory effect on the ATPase activity of ABCG1 have failed.

To examine the sterol-transporter interaction, in the present study, we used CD-complexed sterol compounds. The formulation of these highly hydrophobic molecules facilitates their loading into membranes containing the transporters. Both long and short isoforms of ABCG1 were expressed in the Sf9 insect cell heterologous expression system, and the vanadate-sensitive ATPase activities of membrane preparations from these cells were determined. Both ABCG1 isoforms exhibited relatively high basal ATPase activities as compared to the control, i.e., β-galactosidase-containing membranes (Figure 1 and Table 1). No significant difference between the basal ATPase activities of the long and short ABCG1 isoforms was observed (32.07 ± 0.94 and 34.49 ± 1.64 nmol P_i_/mg protein/min, respectively). Catalytically inactive variants carrying mutations in the Walker A motifs are frequently used as negative controls for functional studies of ABC transporters. This is exemplified by the K433M/K1076M double mutant of ABCB1/MDR1, the K464M/K1250M double mutant of ABCC7/CFTR, and the K86M mutant of ABCG2. As in our previous studies [27,28,33], here we used the catalytic site mutant variant of the long isoform, ABCG1-K124M (G1-LKM), as a negative control. The basal ATPase activity of G1-LKM-containing membrane preparations was not different from that of the β-gal control. Loading the membranes with cholesterol resulted in a substantial (about 1.7-fold) stimulation of the ATPase activities of both wild type (wt) ABCG1 isoforms (Figure 1A,C). In contrast, it had no effect on the ATPase activity of the G1-LKM-containing membranes. Methyl-β-cyclodextrin alone had no effect on the ATPase activity of either the wild type or the inactive mutant form of ABCG1 (Figure 1A). At higher concentrations, a slight inhibition of cholesterol was observed in both G1-L- and G1-S-containing membranes.

In addition to cholesterol, other sterol compounds were investigated. The chemical structures of the studied sterols are shown in Appendix A. Cholesterol precursors, such as desmosterol and lathosterol, stimulated G1-L-mediated ATPase activity about 2-fold and 1.5-fold, respectively (Figure 1A). As cholesterol, desmosterol, and lathosterol inhibited the ATPase activity at higher concentrations. Among the oxidized forms of cholesterol, 7-ketocholestrol exhibited a significant stimulatory effect (2-fold), whereas 25-OH-cholesterol and 27-OH-cholesterol had no effect (Figure 1B). Unlike cholesterol and desmosterol, 7-ketocholestrol did not inhibit the ATPase activity even at the highest studied concentration. Unexpectedly, plant-derived sterol compounds, such as β-sitosterol and campesterol, also efficiently stimulated the ATPase activity of G1-L. None of the studied compounds had an effect on the membranes containing the inactive mutant form of ABCG1. The relative stimulatory effects of the studied sterol compounds were summarized in Table 1. In this case, the highest ATPase activity (peak stimulation) was normalized to the basal ATPase activity. For comparison, the stimulation by rhodamine 123 (rhod123), a previously identified potential ABCG1 substrate [33], was also indicated.

As with the long isoform of ABCG1, the short variant was also stimulated by numerous sterol compounds (Figure 1C,D). The similar basal ATPase activities and rhod123 stimulations (see Table 1) suggest that the two isoforms are functionally expressed in the Sf9 membranes at comparable levels. Although stimulation of cholesterol was not different in G1-L- and G1-S-containing membranes, slight differences with other sterol compounds were observed. Desmosterol, 7-ketocholestrol, and β-sitosterol stimulated G1-S ATPase activity to a significantly lesser extent as compared to that of G1-L. In contrast, 25-OH-cholesterol stimulated only the short isoform and remained inactive in the G1-L-containing membranes. Taken together, these results suggest selective sensitivities to sterols for the two ABCG1 isoforms.

Similar experiments were performed with Sf9 membrane preparations containing the ABCG4 protein. In a previous study, we have shown that ABCG4-containing membranes exhibit a slightly elevated basal ATPase activity over background, which cannot be further stimulated by rhod123 [33]. In that study, over 100 compounds were screened for their interaction with ABCG4, but none of them had a stimulatory effect on the basal ATPase activity of ABCG4. Now we added the sterol compounds in CD complexed form to the G4-containing membranes and found that several of these compounds, including cholesterol, desmosterol, 7-ketocholesterol, 25-OH-cholesterol, and β-sitosterol, were able to stimulate the ATPase activity of ABCG4 (Figure 1E,F). In contrast to the ABCG1 isoforms, ABCG4 was not stimulated by campesterol. It is important to note that the studied proteins (ABCG1 and ABCG4) were untagged, and therefore, different antibodies were used for their immunodetection; thus, their expression levels cannot be compared. The basal ATPase activity of ABCG4 was relatively small but significantly higher than background. The relative stimulations to the basal ATPase activity (fold change) in response to several sterols mainly exhibited a similar pattern to that seen with G1-L (Table 1), with the exception of 25-OH-cholesterol, which stimulated ABCG4 ATPase activity to a similar extent to that seen with G1-S.

### 3.2. Effect of Mutations in the Potential Sterol-Binding Sites in ABCG1 on Its ATPase Activity

The stimulation of ABCG1 ATPase activity by various sterol compounds implies that these molecules interact with the transporter. Since no marked difference was seen between the two ABCG1 isoforms (neither between ABCG1 and ABCG4), and for the reason that the short isoform is the predominant variant in humans, G1-S was used in the subsequent experiments. To identify regions in the ABCG1 protein that interact with the sterols, we performed mutational analyses. Based on sequence analysis, five potential sterol-binding sites were identified, including three CRAC motifs and two SBE sequences. These motifs and an additional site, phenylalanine at position 571, were mutated. The latter is homologous to the position in ABCG2, which has been suggested to play a role in the interaction between the transporter and cholesterol [32] and to serve as a gatekeeper along the transport pathway [4,38]. The positions of these investigated motifs are indicated in Appendix A. One of the leucines and the central tyrosine in the SBE and CRAC motifs, respectively, as well as the phenylalanine at 571, were replaced with alanine by site-directed mutagenesis. The DNA constructs for both the Sf9-baculovirus and the mammalian expression vector were generated.

First, the ATPase activities of the ABCG1 mutants in Sf9 membrane preparations were investigated. Since the expression levels varied to some extent for each variant, the amount of membranes used for the ATPase assay was adjusted accordingly. To verify the comparable levels of ABCG1 variants in the membrane preparations, the samples were analyzed by Western blotting (Figure 2A). The basal ATPase activities of most ABCG1 mutants were comparable with those of the wt ABCG1, except for the Y586A and Y655A CRAC mutants, which exhibited substantially lower basal ATPase activities (Figure 2B and Appendix A). These variants exhibited basal ATPase activities similar to those seen in β-gal-containing membranes (6.04 ± 1.40 nmol P_i_/mg protein/min). Cholesterol stimulated the ATPase activity of the wt transporter about 2-fold, whereas this stimulatory effect of cholesterol was completely abolished in the two CRAC mutants that showed no basal ATPase activities over background (Y586A and Y655A). Interestingly, cholesterol stimulation was abolished in the F571A and L626A mutants, whereas it was reduced to some extent in the L430A variant. The ATPase activity of the Y660A was basically unaffected. Surprisingly, rhod123 stimulation was almost identical to that achieved with cholesterol in all mutants (Figure 2B).

### 3.3. Expression and Apoptotic Effect of ABCG1 Mutants in Mammalian Cells

In addition to Sf9 insect cells, we have expressed the various ABCG1 variants in mammalian cells, i.e., HeLa and HEK293 cell lines. We employed these cells for the reasons that they are well-characterised null cell lines (not expressing endogenous ABCG1 or ABCG4) of human origin and can be transfected with high efficiency. First, their subcellular localisation was investigated by immunofluorescence staining and subsequent confocal microscopy. The wt ABCG1 was predominantly found in the plasma membrane in both cell types (Figure 3), but some intracellular staining was also observed, which is in accordance with our previous finding [27]. Giantin, a Golgi-resident protein, co-localised with wt ABCG1 (Appendix A), suggesting that the intracellular compartment in which the transporter was found is the Golgi complex. The catalytic site mutant variant of the short isoform of ABCG1 (G1KM) exhibited similar subcellular distribution with even more prominent plasma membrane localisation (Figure 3). The L430A and Y660A mutants were also targeted to the plasma membrane in both cell types, whereas the Y586A variant was localised predominantly to the plasma membrane in HEK293 cells but showed diverse (plasma membrane and intracellular) distribution in HeLa cells. The F571 mutant in HeLa cells was mostly found in intracellular compartments, although a faint plasma membrane staining was also observed. Contrary to that seen in HeLa cells, this mutant was predominantly localised to the plasma membrane in HEK293 cells. The L626A and Y655A mutants exhibited intracellular localisation in both cell types.

Previously, we observed that ABCG1 induces apoptosis in cells when expressed at higher levels [27]. Not only forced expression but also induction of the endogenous ABCG1 leads to programmed cell death. Although the exact mechanism of ABCG1-induced apoptosis is unknown, it is certain that this phenomenon is closely connected to the activity of the transporter, since the ATPase-inactive variant of ABCG1 has no apoptotic effect and inhibitors of the ABCG1 function, such as thyroxin or benzamil, completely block the ABCG1-mediated apoptosis [27]. Later, we demonstrated that the closely related transporter ABCG4 also induces apoptosis when overexpressed [27,28]. Moreover, this phenomenon was used to demonstrate that ABCG1 and ABCG4 can form functional heterodimers. In the present study, we also used ABCG1-mediated apoptosis as a functional readout of the transporter. We examined how mutations in CRAC and SBE motifs affect ABCG1-induced apoptosis in HEK293 and HeLa cells. As a negative control, the inactive mutant variant G1KM was used, which does not cause programmed cell death [27]. For comparison, the apoptotic effects of G1-L and G1-LKM were also studied. We found that G1-S induced apoptosis to a similar extent to that observed with G1-L in both cell types, and the catalytic site (KM) mutations abolished their apoptotic effect (Figure 4 and Appendix A). The L430A and L660A substitutions did not affect the ABCG1-induced apoptosis, although the effect of the L660A mutant was slightly but not significantly smaller than that of wt ABCG1. In contrast, the L626A, Y586A, and Y655A mutations diminished ABCG1-associated apoptosis to the level observed in the inactive mutants (G1KM and G1-LKM). The F571 mutant did not induce programmed cell death in HeLa cells but did in HEK293 cells (Figure 4B). It should be noted here that the adherence of HEK293 cells is feeble and sensitive to any perturbation; thus, assessment of apoptosis in these cells is technically challenging and results in substantial deviation. We compared differences in every combination using ANOVA followed by Tukey’s HDS post-hoc test, which demonstrated that there are generally two groups of the variants: (i) those that induce apoptosis to a similar extent to the wild type (L430A, Y660A, and G1-L); and (ii) those that have an apoptotic effect similar to that of the G1KM (F571A, L586A, Y655A, and G1-LKM). The differences within the groups were not significant, but they were significant in comparison with any member of the other group. There was only one exception: the difference between Y655A and Y660A was not significant, indicating that Y660A slightly affects ABCG1’s ability to induce apoptosis.

### 3.4. Positions of Variuos CRAC and SBE Mutation in the ABCG1 Structure

To get an indication of how these mutations affect the function of ABCG1, we highlighted the altered amino acids in the atomistic protein structure (Figure 5). For a better understanding of ABCG1-cholesterol interactions, the cholesterol molecules resolved in the structure were also indicated, and their relative positions to the mutated residues were investigated. L430 was found at the extracellular end of transmembrane helix (TH) 1 at the boundary of the lipid bilayer. In contrast, F571 is located in the center of the protein, above the ’central cavity’ (Figure 5). This phenylalanine, along with F570, seems to form a structure that is analogous to the leucine plug or valve described in the ABCG2 structure [4,38]. The other SBE mutation involves L626, which is positioned in the extracellular loop between TH5 and TH6. Interestingly, its side chain is in close proximity to the so-called reentry helix (also termed the G-loop or TM5b,c), which is highlighted in Figure 5C. Among the central tyrosines in the studied CRAC motifs, Y655 and Y660 are located in the lower (intracellular) half of TH6, close to the C terminus of the protein, while Y586 is positioned in the reentry helix with an outwardly facing side chain (Figure 5C). Interestingly, this latter amino acid is in close proximity to the cholesterol molecules resolved with the ABCG1 structure (Figure 5B,C). It is worth mentioning that none of the other studied mutations were found near any of the resolved cholesterol molecules.

## 4. Discussion

Most previous studies connecting ABCG1 and ABCG4 to HDL-dependent cholesterol removal demonstrated cellular efflux of various sterol compounds, including cholesterol, desmosterol, and 7-ketocholesterol, in the presence of an acceptor molecule, such as BSA, HDL, and cyclodextrin [10,12,14,15,16,17,18,20,21,22,23]. This experimental setup leaves the question open as to whether these sterols are in direct interaction with the transporter or whether their cellular efflux is a downstream event. The stimulation of the ATPase activity of an ABC transporter is widely used as an indicator for substrate-transporter interaction [36,39,40,41,42,43], which has also been applied to the proposed sterol transporters, ABCG1 and ABCG4 [23,33]. Although the highly hydrophobic character of sterols makes experimentation challenging, the cyclodextrin complexation of these compounds allowed us to target them to membranes containing the transporters. Our observation that cholesterol, desmosterol, and 7-ketocholesterol stimulate the ATPase activity of both ABCG1 isoforms and ABCG4 indicates a direct interaction between the transporter and these sterol compounds. The cellular efflux of various sterols, previously reported in connection with these ABCG proteins, and the direct sterol-transporter interaction demonstrated here may implicate that these proteins actually transport these sterol compounds. Our observations are in concert with the previous finding with GFP-tagged ABCG1 purified and reconstituted in liposomes, which was shown to interact with cholesterol and sphingomyelin [23]. Cholesterol was also reported to stimulate the ATPase activity of ABCG2 [36], although to a lesser extent than that which we observed here with the ABCG1 isoforms and ABCG4. This modest stimulation of ABCG2 was, however, attributed to the allosteric effect of cholesterol, which only augments the transport of other molecules [36].

Various sterol compounds have been observed to be accumulated in the brain, retina, and macrophages of Abcg1/Abcg4 and Abcg1/Apoe double knockout mice, but accumulation of phytosterols has not been reported [16,17,25]. Thus, it is unexpected that the plant-derived sterols, such as β-sitosterol and campesterol, interact with ABCG1 and ABCG4. These sterols are re-secreted in the intestine by a transport process mediated by the heterodimer-forming ABCG5/ABCG8 [44,45], which are closest relatives of ABCG1 and ABCG4. Co-immunoprecipitation experiments demonstrated that ABCG1 and ABCG4 can form either homodimers or heterodimers with each other [28], but not with ABCG5 or ABCG8 [10]. Although the ATPase activity assay demonstrates only the interaction between the studied molecule and the transporter, our recent observations raise the possibility that ABCG1 and ABCG4 can transport plant sterols. The sterol recognitions of the two ABCG1 isoforms and ABCG4 were mainly overlapping. However, the G1-L-mediated ATPase activity was not stimulated by 25-OH-cholesterol and was only moderately stimulated by 27-OH-cholesterol and lathosterol, whereas these compounds were fully effective in G1-S. Interestingly, 25- and 27-OH-cholesterol are known LXR agonists, which regulate the transcription of ABCG1 and ABCG4 [46,47]. This transcriptional control of the transporters and the presumed transport of the oxidized cholesterols implicate a negative feedback loop in a “transporter on demand” regulatory mechanism. Moreover, the differential interaction between different hydroxycholesterols and various ABCG1 isoforms makes a complex, fine-tuned control of sterol metabolism possible.

These observations, in line with the results of previous studies, strongly suggest a direct interaction between ABCG proteins and sterol compounds. Therefore, our second set of experiments aimed at mapping the putative sterol binding sites in ABCG1 using a mutational analysis approach. Interestingly, rhod123 stimulation of the ATPase activity of ABCG1 mutants showed an identical pattern to that seen with cholesterol, suggesting that the interacting sites for rhod123 and cholesterol are related (if not identical). Among the studied CRAC and SBE mutations, L430A and Y660A did not affect either the ATPase activity or the apoptotic effect of ABCG1. Moreover, the cellular localisation of the protein carrying one of these mutations also remained unaltered. In line with our observation, leucine substitution of Y672 in the long ABCG1, which corresponds to Y660 in G1-S, has no effect on the cellular cholesterol efflux and the half-life of the transporter [48]. The observations that neither L430A nor Y660A had functional consequences can easily be perceived on the basis of their positions in the protein structure. As mentioned earlier, L430 is situated at a helix-loop border, and its side chain is not involved in extensive residue interactions. Therefore, shortening the side chain by the L-A substitution in such a position is not expected to have a drastic effect. The Y660 residue is located close to the C-terminus of the protein, linked to the nascent poly-peptide chain at the very final stage of protein synthesis. Moreover, the outwardly facing position of the side chain of Y660, similar to that of L430, makes its contribution to intramolecular interactions unlikely. A previous computational study predicted another residue in this CRAC motif, namely R675 in G1-L (corresponding to R663 in G1-S), to be important for cholesterol binding [49]. Our recent experimental observation with the Y660A mutant does not support the significance of this CRAC motif.

Another group of mutations is represented by L626A and Y655, which have severe effects on the function and cellular localisation of ABCG1. These mutations abolished both cholesterol- and rhod123-stimulated ATPase activities in ABCG1. Moreover, the Y655A mutation also caused abrogation of basal ATPase activity. In mammalian cells, both L626A and Y655 were localised in intracellular compartments and did not induce apoptosis. In accordance with our finding, Sharpe et al. reported that the mutation at Y667 in G1-L, which corresponds to Y655 in G1-S, diminished cellular cholesterol efflux and the protein half-life [48]. The close proximity of L626 to the reentry helix makes this residue interact with hydrophobic amino acids in the reentry helix. The interacting counterpart in the static structure is I598, and the neighbouring hydrophobic L599 residue may also interact with L626 under physiological conditions due to dynamics. Similarly, Y655 may be involved in intramolecular interactions. The side chain of this tyrosine is directed toward TH3, right at the position where Y487 is located. Assuming an aromatic-aromatic interaction between these tyrosines is also reasonable. Taking our experimental observations and structural considerations into account, L626A and Y655A seem to affect protein structure substantially, which results in both trafficking and functional defects. Although the exact mechanism of ABCG1-mediated apoptosis is unknown, it seems to require localisation of the transporter at the plasma membrane. Thus, conclusions on the functionality of these variants cannot be drawn on the basis of their apoptotic effects. Since L626 and Y655 are located in the transmembrane domain of ABCG1, and the ATPase activity, which is carried out by the NBDs, is abolished, it is plausible to assume that the allosteric communication between TMD and NBD is abrogated by these mutations. It should also be noted that Y667 in G1-L (corresponding to Y655 in G1-S) has been proposed as a putative cholesterol-binding site [48], but our results cannot either support or exclude this possibility. It is noteworthy that despite, the mentioned abnormalities, the protein carrying either of these mutations is expressed at a normal level, implying that the structural defects caused by these mutations are not so severe that they bring about immediate degradation. In addition, the basal ATPase activity was preserved in the L626A mutant, suggesting that this mutation does not completely abolish the function of ABCG1 but abrogates its ability to be stimulated with cholesterol. This observation suggests that L626 may contribute to the cholesterol sensing of ABCG1.

Similar to L626A, substitution of F571 for alanine results in preserved basal ATPase activity but eradicates cholesterol (and rhod123) stimulation. In mammalian cells, F571 exhibited intermediate behaviour. Its apoptotic effect was diminished (especially in HeLa cells), but not completely abolished. It should be noted that the cellular processing of this mutant was also affected in HeLa cells. As mentioned earlier, this phenylalanine, along with F570 and the corresponding phenylalanines in the other half of the homodimer, are in an arrangement similar to the so-called leucine valve in ABCG2, which is formed by four leucines (L554, L555, L554′, and L555′). A gatekeeping function was attributed to this structure. Based on the structural homology and hydrophobicity of these residues, it seems reasonable to assume a similar role for these four phenylalanines in ABCG1. This notion is supported by the positions of these residues in the open and closed experimental structures of the transporter [8]. Altering the relatively large side chain of F571 to a small one (alanine) may interfere with this gatekeeping function, resulting in a transport-incompetent conformation. However, the observations that the basal ATPase activity of this mutant remained unaffected and that its apoptotic effect was also seen when the transporter reached the plasma membrane suggest that the transport function is presumably preserved. Admittedly, the ABCG1-driven transport is not assessed in our experiments directly, but both ATPase activity and apoptosis mediated by ABCG1 are closely linked to the transport function [27,34]. Importantly, the definite loss of cholesterol stimulation of ATPase activity of the F571A mutant (Figure 2) implicates that this residue may somehow be connected to cholesterol sensing of ABCG1. This mutation likely decouples an allosteric pathway for propagation of dynamic alterations and does not disrupt a direct interaction with cholesterol since this residue is located in the loop between TM5a and TM5b of the reentry helix and is not exposed to the membrane bilayer (Figure 5C).

Intriguing is the position of Y586, as it is located in the reentry helix in close proximity to cholesterol molecules resolved in the structure. This localisation raises the possibility that the CRAC motif in the reentry helix (aa581-591) is crucial for cholesterol sensing. When Y586 is mutated to alanine, both ABCG1-mediated ATPase activity and the apoptotic effect are diminished. Since Y586 is close to F571 in the sequence, namely, F571 is located in a loop preceding the CRAC motif in the reentry helix, the central tyrosine of which is Y586, the propagation of dynamic effects between these two regions via the polypeptide backbone is highly possible when a cholesterol molecule is bound. Moreover, the steric proximity and presumed hydrophobic interaction of L626, whose mutation resulted in the loss of cholesterol sensing, further support the essential role of the reentry helix. These observations envisage a model in which the CRAC motif in the reentry helix is the sensor for cholesterol, which in turn allosterically affects the gatekeeper of the transporter located at the top of the central cavity. However, verification of this model requires further experimental and computational studies.

An intriguing question is whether similar structural arrangements can be found in the closely related transporters, such as ABCG4, ABCG2, and ABCG5/ABCG8. While the multi-specific transport of ABCG2 is allosterically modulated by cholesterol [31,36], the ABCG5/ABCG8 heterodimer is known to mediate the transport of plant-derived sterols [44,45]. In addition, our recent observations indicate direct interactions of sterols with ABCG4 (Figure 1E,F). Thus, we compared the structure of ABCG1 with the recently published model structure of ABCG4 (AlphaFold-Multimer prediction at https://3dbeacon.hegelab.org, accessed on 4 July 2022) [9], as well as the known cryo-EM structures of ABCG2 [3] and ABCG5/ABCG8 [2]. This comparative structure analysis demonstrated that the side chain of the central tyrosine of the reentry helix CRAC motif in ABCG4, ABCG2, and ABCG5 faces outwardly similar to that seen with Y586 in ABCG1 (Appendix A). In contrast, ABCG8 has no CRAC motif in this region. The close proximity of these tyrosines to the gatekeeper structure (e.g., the leucine valve in ABCG2) makes the propagation of dynamic effects upon cholesterol binding conceivable.

In conclusion, we demonstrated here a direct and selective interaction of numerous sterol derivatives with ABCG1 isoforms and ABCG4, which observation provides firm support for the concept that these members of the ABCG subfamily actually transport sterol molecules. Mapping the potential sterol-binding motifs in ABCG1 by mutational analysis and its interpretation in the context of the atomic structure helps us better understand the transporter-sterol interactions. Based on our experimental observations and structural considerations, we propose that the reentry helix (G-loop, TH5b,c) in ABCG1 plays a pivotal role in this interaction. A deeper understanding of the molecular details of a physiologically relevant transporter protein, such as ABCG1, may contribute to the identification of new drug targets for diseases associated with this transporter, such as atherosclerosis, obesity, adipocyte lipid storage, type 2 diabetes, and age-related macular degeneration.

## Figures and Tables

**Figure 1 ijms-23-13744-f001:**
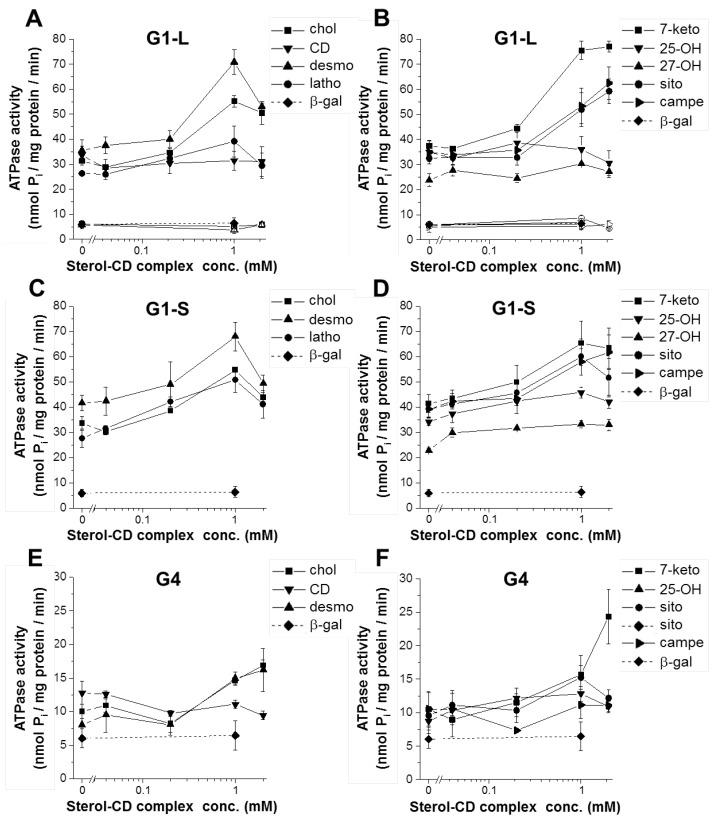
Interaction of various sterol compounds with ABCG1 isoforms and ABCG4. Dose response curves show the vanadate-sensitive ATPase activity of membrane preparations containing the long (**A**,**B**) or the short (**C**,**D**) isoforms of ABCG1, or ABCG4 (**E**,**F**). Open symbols in Panels (**A**,**B**) indicate the activities obtained with membranes containing the catalytic site mutant variant of the long isoform of ABCG1. Membranes containing β-galactosidase (β-gal) served as a negative control. The samples were pre-treated with the indicated sterol compounds at various concentrations or the vehicle (methyl-β-cyclodextrin—CD). chol—cholesterol; desmo—desmosterol; latho—lathosterol; 7-keto—7-ketocholesterol; 25-OH—25-hydroxycholesterol; 27-OH—27-hydroxycholesterol; sito—β-sitosterol; campe—campesterol.

**Figure 2 ijms-23-13744-f002:**
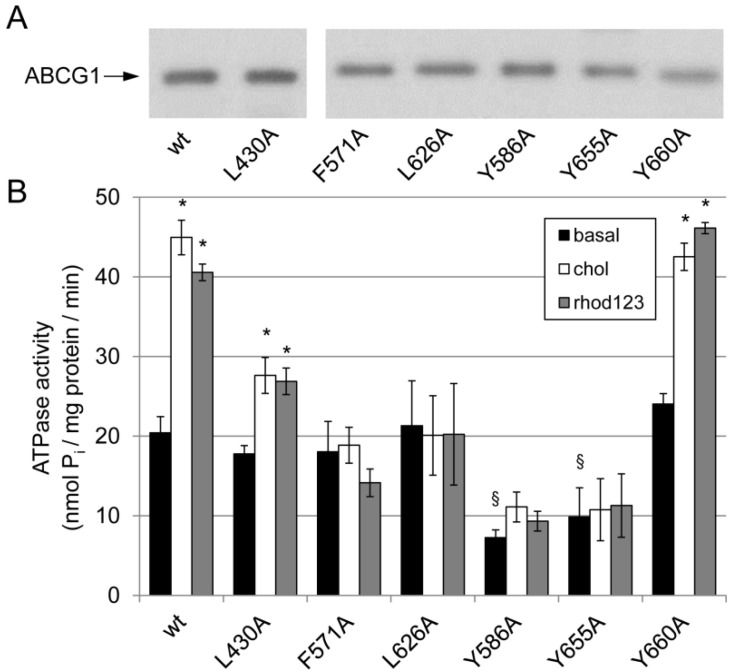
Expression and ATPase activity of ABCG1 mutant variants. (**A**) The amount of ABCG1 variants expressed in Sf9 cells was adjusted to comparable levels on the basis of Western blot analysis. (**B**) Section symbols (§) indicate significantly reduced basal ATPase activity as compared to wild type (wt), whereas asterisks (*) mark significant differences as compared to the basal activity (*p* < 0.05).

**Figure 3 ijms-23-13744-f003:**
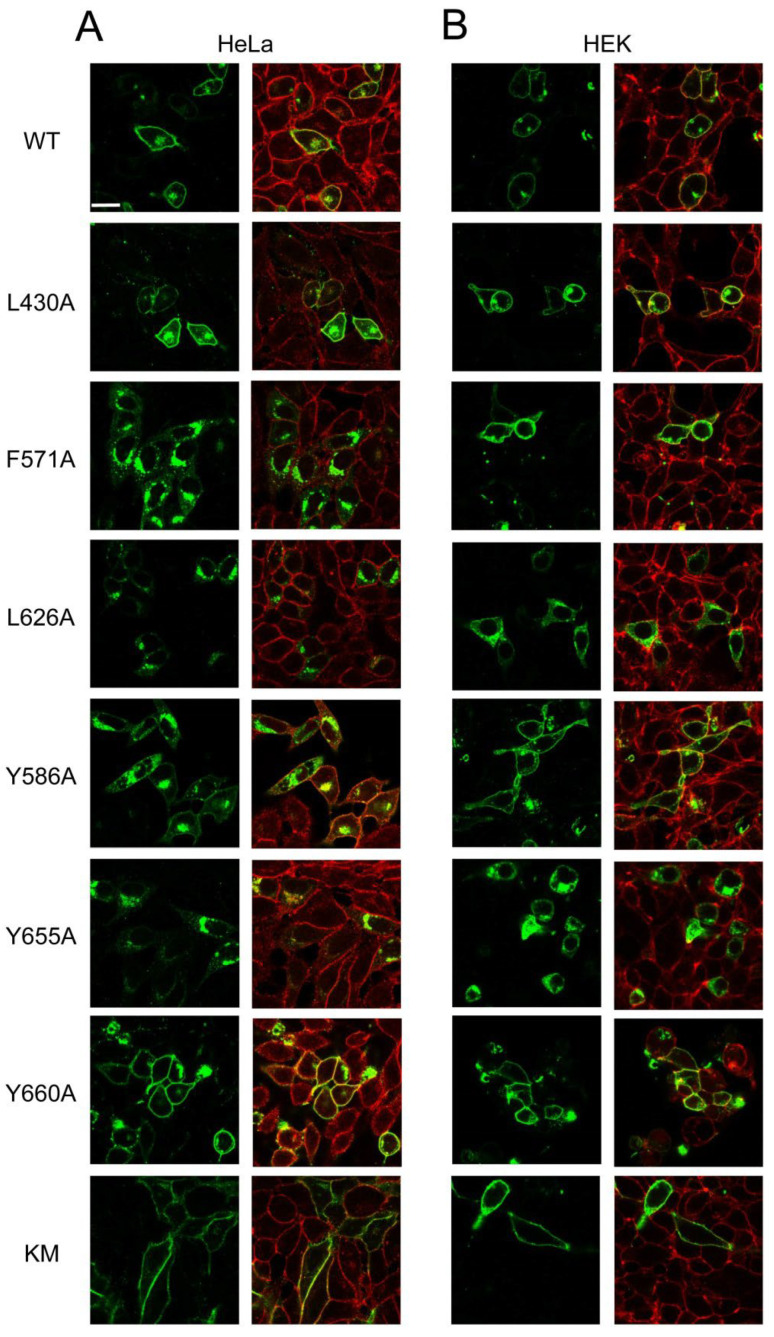
Subcellular localization of ABCG1 mutant variants in mammalian cells. Variants of ABCG1-S were expressed in HeLa (**A**) and HEK293 (**B**) cells, immunostained for ABCG1 (green) and cadherin (red), and imaged by confocal microscopy.

**Figure 4 ijms-23-13744-f004:**
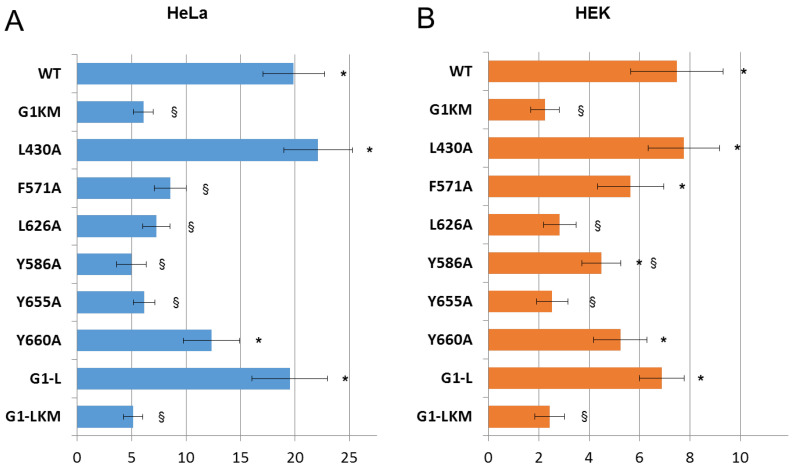
Apoptosis induced by various ABCG1 mutants. Variants of ABCG1-S were expressed in HeLa (**A**) and HEK293 (**B**) cells, and the fraction of the apoptotic cells was determined by Annexin V binding. For comparison, the apoptotic effects of the long form (G1-L) and its inactive mutant variant (G1-LKM) are also shown. Section marks (§) and asterisks (*) indicate significant differences as compared to wt ABCG1 (WT) and G1KM, respectively (*p* < 0.05).

**Figure 5 ijms-23-13744-f005:**
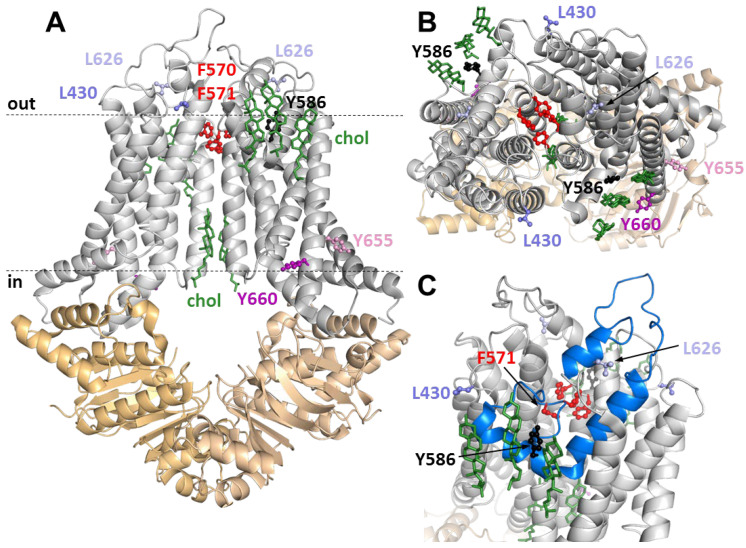
The homodimeric structure of ABCG1. Side view (**A**) and top view from the extracellular space (**B**) of ABCG1/ABCG1 (PDBID: 7r8d, G1-S). NBDs and TMDs are coloured in orange and gray, respectively. Cholesterol molecules resolved in the structure are shown by green sticks. F570 and F571, separating the main intracellular and extracellular substrate binding cavities, are red. The residues mutated in this study are shown in different colours, represented by sticks and balls, and labeled by residue numbers. (**C**) The residue Y586 (black), which is located in TM5b of the reentry G-loop (blue), interacts with one of the cholesterol molecules. Parts of the extracellular loop until the helix, including L626, are also coloured in blue. Dashed lines indicate the approximate location of the lipid bilayer and the hydrophobic membrane-embedded region of the protein.

**Table 1 ijms-23-13744-t001:** Comparison of basal and stimulated ATPase activities of ABCG1 isoforms and ABCG4. Basal ATPase activities of the long (G1-L) and short (G1-S), as well as that of ABCG4 (G4), are shown in the second row (highlighted with orange background), whereas relative stimulations (peak ATPase activities normalised to the basal activities) are indicated in the subsequent rows (under the dark green header).

	G1-L	G1-S	G4
basal activity (nmol P_i_/mg protein/min)	32.07 ± 0.94 ^§^	33.15 ± 1.51 ^§^	10.78 ± 0.62 ^§^
peak stimulation (fold change)			
cyclodextrin	0.94 ± 0.06	1.02 ± 0.00	0.89 ± 0.09
cholesterol	1.76 ± 0.06 *	1.64 ± 0.11 *	1.72 ± 0.20 *
desmosterol	2.02 ± 0.12 *	1.63 ± 0.10 *^†^	1.91 ± 0.25 *
lathosterol	1.51 ± 0.31	1.92 ± 0.22 *	n.d.
7-ketocholesterol	2.02 ± 0.06 *	1.56 ± 0.12 *^†^	2.18 ± 0.25 *
25-OH-cholesterol	1.02 ± 0.06	1.34 ± 0.07 *^†^	1.55 ± 0.21 *^†^
27-OH-cholesterol	1.31 ± 0.17	1.50 ± 0.12 *	n.d.
β-sitosterol	1.81 ± 0.09 *	1.52 ± 0.09 *^†^	1.63 ± 0.25 *
campesterol	1.84 ± 0.26 *	1.57 ± 0.07 *	1.24 ± 0.42
rhodamine 123	1.91 ± 0.06 *	1.71 ± 0.09 *	1.08 ± 0.00 ^†^

dark green—stimulation of all three ABCG proteins; light green—stimulation of both isoforms of ABCG1; yellowish green—stimulation of G1-S only; yellow—stimulation of G1-S and G4. ^§^ significant difference as compared to the activity of β-gal-containing membranes (6.04 ± 1.40 nmol P_i_/mg/min); * significant difference from 1, i.e., peak stimulations are compared to basal activities; ^†^ significant difference as compared to the peak stimulation of G1-L; n.d.—not determined.

## Data Availability

Not applicable.

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
