# Peer review of "The Reentry Helix Is Potentially Involved in Cholesterol Sensing of the ABCG1 Transporter Protein"

_ijms, 2022, doi:10.3390/ijms232213744_

Round 1

Reviewer 1 Report

ABCG1 and ABCG4 are members of the G subfamily of ATP-binding cassette transporters and details of their interaction with sterol substrates remain elusive. In this work, the authors investigated the effects of several sterol compounds on these proteins as well as attempted to identify the motifs responsible for this interaction. Using a combination of tools involving ATP-ase assay in isolated membranes (of SF9 insect cells), mutational analysis of the proposed Cholesterol Recognition Amino acid Consensus (CRAC) and Sterol Binding Element (SBE) motifs, cellular localization and apoptosis detection experiments, the authors propose an essential role for the reentry helix in cholesterol sensing. A major limitation of the study is the ambiguity in the interpretation of the results from the ATP-ase assays and mutation analysis towards function, dynamics and the stability of the transporter, in particular in the absence of complementary data. 

1.     The observation that different sterols (cholesterol, desmosterol, and 7-ketocholesterol, etc.,) stimulate the ATPase activity of both ABCG1 isoforms and ABCG4 indicates an interaction between the transporter and these sterol compounds, Yet, this could be an indirect effect of transporter-sterol interaction and might not confirm them as the real substrate(s). 

2.     Mutations L626A and Y655A abolished both cholesterol and rhod123-stimulated ATPase activities in ABCG1 (Y655A mutation also caused abrogated basal ATPase activity). These variants are expressed at normal levels, yet their effects are attributed to reduced protein stability. Also, ATP-ase activity, which is the major experimental readout occurs at the NBDs; thus, these mutations may also affect the TMD-NBD cross-talk or the associated kinetics, etc. (among the several other possibilities).

3.     The Y586A mutation located on the reentry helix diminished the ABCG1-mediated ATPase activity and apoptotic effect. It is an interesting suggestion that this CRAC motif could be a sensor for cholesterol, which would intern allosterically affect the phenylalanine gate-keeper at the top of the cavity. As the authors themselves suggested, further experimental and computational studies are required to elucidate the exact role of Y586.

4.     For the apoptosis assay, the authors did not interpret the large difference observed between the mutants. 

5.     A transport assay or a transporter-sterol binding/release experiment would be helpful/required to reliably interpret the current mutational analysis to the function (sterol transport). Further, the authors might also consider a combined mutational analysis (for e.g., involving residues they propose to interact).

Minor comments

1.     Please briefly mention how/why ABCG1 may promote apoptosis.

2.     Please briefly explain the K124M inactive mutant. In many ABC transporters, this corresponds to an ExxxQ mutation.

3.     The authors present very limited supporting information. Please consider including the raw data for ATP-ase and apoptosis assays and comparative structural analysis (including that for (CRAC and SBE) motifs).

4.     Figure 1 – please check for the font size

Author Response

We are grateful to the reviewer for the thorough revision of our manuscript and the valuable suggestions.

  1. The observation that different sterols (cholesterol, desmosterol, and 7-ketocholesterol, etc.,) stimulate the ATPase activity of both ABCG1 isoforms and ABCG4 indicates an interaction between the transporter and these sterol compounds, Yet, this could be an indirect effect of transporter-sterol interaction and might not confirm them as the real substrate(s).

We agree with the reviewer that stimulation of ATPase activity, literally taken, indicates only the interaction between the sterol compound and the transporter. However, transported substrates regularly stimulate the ATPase activity of ABC transporters, such as ABCB1/MDR1, ABCC1/MRP1, ABCG2/BCRP, etc. Therefore, assessment of the ATPase activity stimulation is commonly accepted and widely used method for screening for ABC transporter substrates. Although we tried not to overinterpret these results in our manuscript, in the revised version, we refined our statements in connection with ATPase activity stimulation, and provided more details on advantages and the limitations of the method.

  1. Mutations L626A and Y655A abolished both cholesterol and rhod123-stimulated ATPase activities in ABCG1 (Y655A mutation also caused abrogated basal ATPase activity). These variants are expressed at normal levels, yet their effects are attributed to reduced protein stability. Also, ATP-ase activity, which is the major experimental readout occurs at the NBDs; thus, these mutations may also affect the TMD-NBD cross-talk or the associated kinetics, etc. (among the several other possibilities).

We greatly appreciate this notion of the reviewer. Although a previous work (Sharpe et al. 2015) demonstrated reduced half-life for Y667 in G1-L (which corresponds to Y655 in G1-S), our experiments are, indeed, not suitable for making conclusion on protein stability, since they are steady-state and not kinetic measurements. In the revised manuscript, we rephrased our statements on these mutants. The mislocalization and loss of function observed with these variants suggest that protein structure severely affected, which may result in abrogation of allosteric communication between TMD and NBD.

  1. The Y586A mutation located on the reentry helix diminished the ABCG1-mediated ATPase activity and apoptotic effect. It is an interesting suggestion that this CRAC motif could be a sensor for cholesterol, which would intern allosterically affect the phenylalanine gate-keeper at the top of the cavity. As the authors themselves suggested, further experimental and computational studies are required to elucidate the exact role of Y586.

In the present work, we performed a general screen for the potential cholesterol binding sites in ABCG1. As a result of this, Y586 emerged as a potentially essential residue (or part of an essential region) for cholesterol sensing, which was rather surprising, since membrane topology predicted an extracellular localization for this residue. However, homology modeling and the recently published ABCG1 protein structure, as well as cholesterol molecules in the proximity of Y586in the cryo-EM structure made our results interpretable. This led us to suggest that the reentry helix is crucial for cholesterol sensing, and to envisage the model we proposed in our manuscript (the putative dynamic effect between this region and F571, the “gate-keeper”). A detailed mutational analysis and computational study focusing on only this region of ABCG1 may elucidate the molecular interaction of cholesterol-sensing and propagation of information within the protein. However, these analyses are beyond the scope of the recent study, and can be a part of a follow up study (please also see our response to point 5).

  1. For the apoptosis assay, the authors did not interpret the large difference observed between the mutants.

We primarily focused on whether a mutation diminishes the ABCG1-mediated apoptosis or leaves it unaffected; therefore, we compared a given mutant to the wild type (or to the inactive form). Upon the reviewer’s concern, we performed ANOVA followed by Tukey’s HDS post-hoc test, which demonstrates whether differences are significant or not in every possible relation. We found that there are generally two groups of the variants: i) those that induce apoptosis to similar extent to the wild type (L430A, Y660A, and G1-L) as well as ii) those that have apoptotic effect similar to that of the G1KM (F571A, L586A, Y655A, and G1-LKM). The differences within the groups were not significant, but they were significant in comparison with any member of the other group. There was only one exemption: the difference between Y655A and Y660A was not significant, indicating that Y660A slightly affects ABCG1’s ability to induce apoptosis. We added the results of this analysis to the revised manuscript.

  1. A transport assay or a transporter-sterol binding/release experiment would be helpful/required to reliably interpret the current mutational analysis to the function (sterol transport). Further, the authors might also consider a combined mutational analysis (for e.g., involving residues they propose to interact).

We agree with the reviewer that a more detailed analysis of this region, which includes other single mutants and combined mutations, as well as binding assays and direct transport experiments, would help to understand the molecular details of sterol-transporter interactions better. However, the time requirement for these experiments is far beyond the given timeframe for the revision (10 days). As discussed in the answer to comment 3, the suggested analysis can be performed in the future, and be published in a separate, follow-up paper.

Minor comments:

  1. Please briefly mention how/why ABCG1 may promote apoptosis.

We provided more details on the ABCG1-medied apoptosis in the revised manuscript.

  1. Please briefly explain the K124M inactive mutant. In many ABC transporters, this corresponds to an ExxxQ mutation.

K124M is a mutation in the Walker A motif of the ATP binding site, which leads to abrogated ATPase and transport activity. The mutated lysine can be found also in other ABC transporters, and the KM mutants are commonly used as negative control, such as the K433M/K1076M double mutant of ABCB1/MDR1, the K464M/K1250M variant in ABCC7/CFTR, and the K86M mutant of ABCG2. EQ mutants of various ABC transporters are also used as ATPase inactive variants, but they carry mutation in the Walker B motifs. This is exemplified by the E556Q/E1201Q double mutant of ABCB1/MDR1 and the E558Q/E1207Q double mutant of ABCB11/BSEP. In the revised manuscript, we discuss this issue in more detail, citing the relevant references, where we previously used ABCG1-K124M (Cserepes et al, Seres et al, Hegyi et al).

  1. The authors present very limited supporting information. Please consider including the raw data for ATP-ase and apoptosis assays and comparative structural analysis (including that for (CRAC and SBE) motifs).

We are grateful to the reviewer for this suggestion. We included a comparative structural analysis as a Supplementary Figure (Supplementary Fig. 4). Also the tables containing the raw data for ATPase and apoptosis assays are presented (Supplementary Tables 1 and 2).

  1. Figure 1 – please check for the font size

In the revised manuscript, we improved the lettering in Figures 1 and 5.

We hope that our responses to the reviewer are satisfactory and the improvements implemented in the manuscript make it now acceptable for publication.

Reviewer 2 Report

This manuscript is a well written account of an important area of investigation. Apart from correcting a few typographical errors, the English is of good quality. This article is recommended for publication after consideration of several points given below.

1. Structures of steroids. The inclusion of the structures of the steroids used in this study would be helpful. If space is a problem, the structures could be included in the Supplementary Section. Alternatively, partial structures could be considered.

2. Figures and Table. The lettering in Figure 1 is far too small and some adjustments need to be made. Figure 2, on the other hand, is very clear as is the use of different colors in Table 1. In Figure 5, the lettering should be larger.

3. HEK cells. The authors should indicate why HEK cells were chosen.

Author Response

This manuscript is a well written account of an important area of investigation. Apart from correcting a few typographical errors, the English is of good quality. This article is recommended for publication after consideration of several points given below.

We thank the reviewer for his/her valuable suggestions and the positive evaluation of our manuscript.

  1. Structures of steroids. The inclusion of the structures of the steroids used in this study would be helpful. If space is a problem, the structures could be included in the Supplementary Section. Alternatively, partial structures could be considered.

The structures of steroids are included as a Supplementary Figure (Supplementary Fig. 1) in the revised manuscript.

  1. Figures and Table. The lettering in Figure 1 is far too small and some adjustments need to be made. Figure 2, on the other hand, is very clear as is the use of different colors in Table 1. In Figure 5, the lettering should be larger.

We improved the lettering in Figures 1 and 5. Thanks for the suggestions and the appreciation.

  1. HEK cells. The authors should indicate why HEK cells were chosen.

In the revised manuscript, we added reasoning for the choice of cell lines. Briefly, we used null cell lines (not expressing ABCG1 or ABCG4) of human origin. HEK cells are well characterized and easy to work with. They are transfectable with high efficacy and we applied them in our previous studies on ABCG1 and ABCG4 (see Seres et al. and Hegyi et al.). However, in our current study, we performed the experiments not only with HEK cells but also with HeLa cells to exclude cell type-specific incidents.

We hope that recent changes are satisfactory and our manuscript will be acceptable for publication.

Round 2

Reviewer 1 Report

The authors have faithfully responded to my questions/comments. The authors have interpreted their functional and localization readouts with sufficient precaution. They provided more data in the supporting information and their major conclusions/interpretations are supported by the presented data.